

# Eye Disease Net: an algorithmic model for rapid diagnosis of diseases

Fangyuan Liu[1], Bo Qin[1,2,3] and Fengqi Jiang[1]

[1] The Second Clinical Medical College, Jinan University, Shenzhen, China
[2] Shenzhen Aier Eye Hospital, Aier Eye Hospital, Jinan University, Shenzhen, China
[3] Shenzhen Aier Ophthalmic Technology Institute, Shenzhen, China

## ABSTRACT

With the development of science and technology and the improvement of the quality of life, ophthalmic diseases have become one of the major disorders that affect the quality of life of people. In view of this, we propose a new method of ophthalmic disease classification, ED-Net (Eye Disease Classification Net), which is composed of the ED_Resnet model and ED_Xception model, and we compare our ED-Net method with classical classification algorithms, transformer algorithm, more advanced image classification algorithms and ophthalmic disease classification algorithms. We propose the ED_Resnet module and ED_Xception module and reconstruct these two modules into a new image classification algorithm ED-Net, and compared them with classical classification algorithms, transformer algorithms, more advanced image classification algorithms and eye disease classification algorithms.

## INTRODUCTION

As international education continues to increase and factors such as the social environment continue to change, so does the environment in which the eyes are used. As a result, the risk of ophthalmic disease has also increased. Traditionally, ophthalmic diseases are identified manually by the physician, while the category of disease is determined according to the physician's experience and pathology records (*Ting et al., 2019*), a method that obviously requires a high level of expertise from the physician, while being slow and prone to the problem of misdiagnosis. In view of this, *Panozzo et al. (2020)* proposed a method for grading diabetic macular lesions using spectral sweep technique, which enabled rapid identification of diabetic macular lesions; *Frame et al. (1998)* used the method of convolutional neural network (CNN) (*Alzubaidi et al., 2021*) for the detection of microaneurysms in ophthalmic fluorescein angiograms, which has better detection performance; *Vermeer et al. (2011)* uses OCT (Optical Coherence Tomography) images of the retinal layer to automatically segment the pixels, allowing for more efficient detection of variations in the retinal layer; *Wang et al. (2017)* uses a machine learning approach to ophthalmic images, in specific cases, to assist physicians. *Wang et al. (2017)* uses machine learning methods on ophthalmic images to assist doctors in the diagnosis of ophthalmic diseases in specific situations. All of these methods have improved the efficiency of diagnosis to a certain extent, but they are all disease-specific and do not play a key role in

Corresponding author
Bo Qin, qinbozf@126.com

the early stages of a patient's condition, nor do they allow for the classification of the main ophthalmic diseases. In view of this, we propose a new classification algorithm, ED-Net (Eye Disease Net), which can assist the physician in identifying the patient's major ophthalmic disease in real time at an early stage, helping the physician to treat the patient better and faster. With reference to the main effects and frequency of ophthalmic diseases, we used seven categories of ophthalmic diseases, namely Bulging_Eyes, Cataracts, Crossed_Eyes, Diabetic_retinopathy, Glaucoma, Uveitis and Normal.

Our contributions are shown below, respectively.

(1) A partial dataset of ophthalmic diseases was collected through the Shenzhen Aier Eye Hospital Affiliated to Jinan University and merged with two Kaggle Competition datasets to provide a completely new dataset of ophthalmic diseases.

(2) The ED_Resnet model was optimized and proposed based on the residual module of the ResNet network.

(3) Optimising and proposing the ED_Xception model based on the Xception network module.

(4) The ED_Resnet model and ED_Xception model are stacked to obtain a new ophthalmic image classification model, ED-Net.

## RELATED WORK

With the impressive achievements of deep learning (*Shorten & Khoshgoftaar, 2019*) in numerous fields, such as: image classification (*Naranjo-Torres et al., 2020*), target detection (*Zou et al., 2019*) and image segmentation (*Shervin et al., 2022*), CNN (*Alzubaidi et al., 2021*) and transformer (*Han et al., 2021*) models have gained the favour of many research scholars. CNN and transformer models are made to distinguish and identify different types of image information by continuously extracting features from the image information.

The structural development of CNN was originated by Alexnet (*Lu, Wang & Zhang, 2021*), whose construction consists of an input layer, a convolutional layer (Conv), a pooling layer (average pooling (*Stergiou & Poppe, 2022*), max pooling (*Zhang et al., 2021*), overlapping pooling (*Jeewandara et al., 2021*), Spatial Pyramid pooling (*Huang et al., 2020*)), an activation function layer, a fully connected layer and the output layer. The most used pooling layers are average pooling and max pooling; the most used output layer is the softmax function (*Niklaus & Liu, 2020*).

The maximum pooling calculation is shown in Fig. 1, where four different colours are used to represent different regions, and the calculation is carried out in steps of 2, resulting in a feature map of [5,10,9,7]. The principle of maximum pooling is to take the maximum value in the same coloured region as the size of the calculated pixel value.

The average pooling is calculated as shown in Fig. 2. We select the same area values and then take the average of [2,2,3,5] as the pixel value after the average pooling, the average pooled pixel value after the average pooling is [3,4,6,4].

The softmax function is calculated as shown in Eq. (1). Softmax works by assigning a probability value $P(z_i)$ to each output category, and $P(z_i)$ indicates the probability size of each output category, rather than just determining a maximum value.

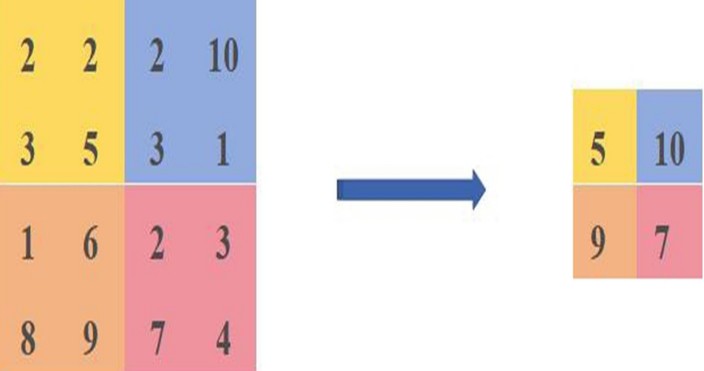

**Figure 1 Diagram of the maximum pooling calculation method.**

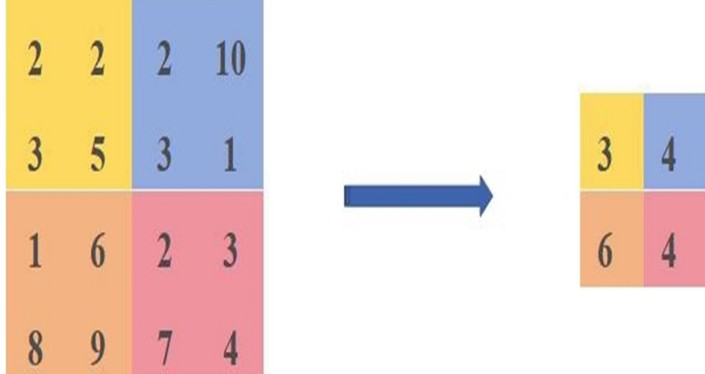

**Figure 2 Diagram of the average pooling calculation method.**

$$P(z_i) = \frac{e^{h(z_i)}}{\sum_{j=1}^{j} e^{z_j}} \tag{1}$$

where $z_i$ is denoted as the output value of the ith i-node and j represents the number of output nodes, which is the number of classifications in the classification. By using the Softmax function the output values of the polyphenolic classes can be converted to probability values of size between [0,1], which sum to 1.

The transformer model is a model structure proposed by researchers at the bottleneck stage of CNN development. The transformer consists of two parts: encoder and decoder, where we use six encoders as encoders and six decoders as decoders. The network structure is shown in Fig. 3.

## OUR METHOD

Our algorithm was built from zero, and the different classification party methods were repeatedly elaborated and optimised to obtain two new modules: the ED_Resnet model

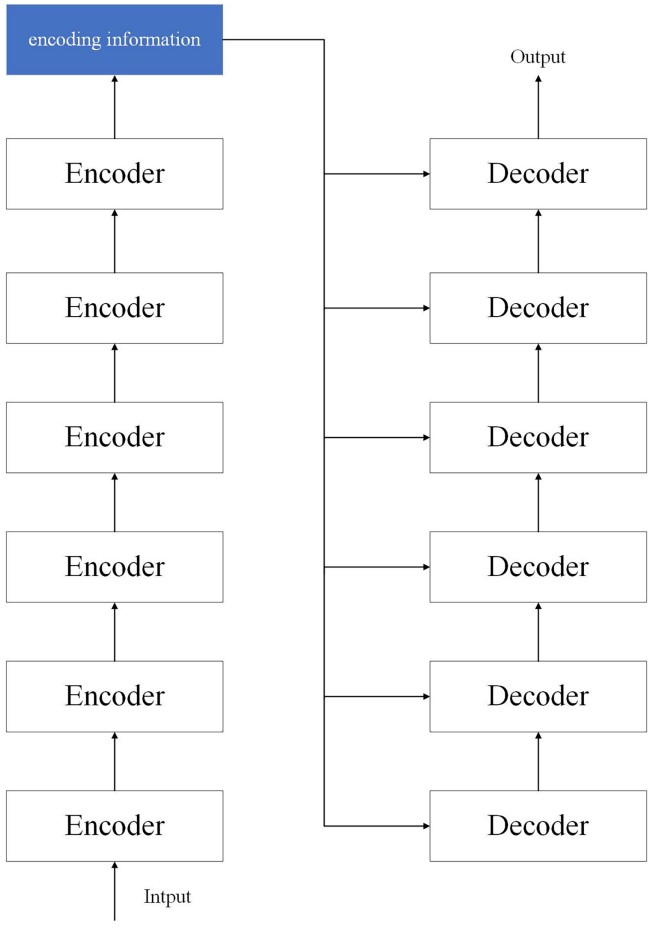

**Figure 3  Structure of the transformer model.**     

and the ED_Xception model. A new ophthalmic disease classification model, ED-Net, was developed.

## ED_Resnet module

The original ResNet (*Wightman, Touvron & Jégou, 2021*) residual residual convolution module and our ED_Resnet convolution module are shown in Fig. 4, respectively. We take inspiration from the figure in (a) and design a new module (b). Our module consists of three convolutions, a $1 \times 1$, a $5 \times 5$ depth convolution and a $1 \times 1$ convolution, while the input image values are directly concatenated with the output of the feature map after three convolutions by jumping to understand the output.

## ED_Xception module

As shown in Fig. 5, we refer to the Xception (*Chollet, 2017*) network architecture, as shown in Fig. 5A, and experimentally optimize Xception's backbone by using $5 \times 5$ and $7 \times 7$ convolutions instead of the original $3 \times 3$ convolutions. Although there is a slight increase in computational effort, we finally adopt the network structure diagram in Fig. 5B during the experiment.

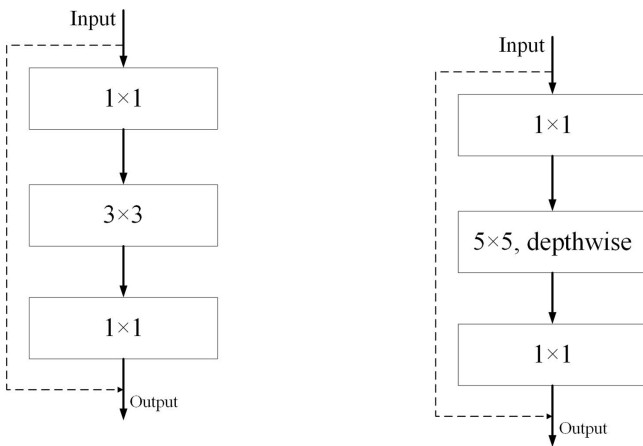

**Figure 4  Diagram of ResNet and ED-Net convolution modules.**

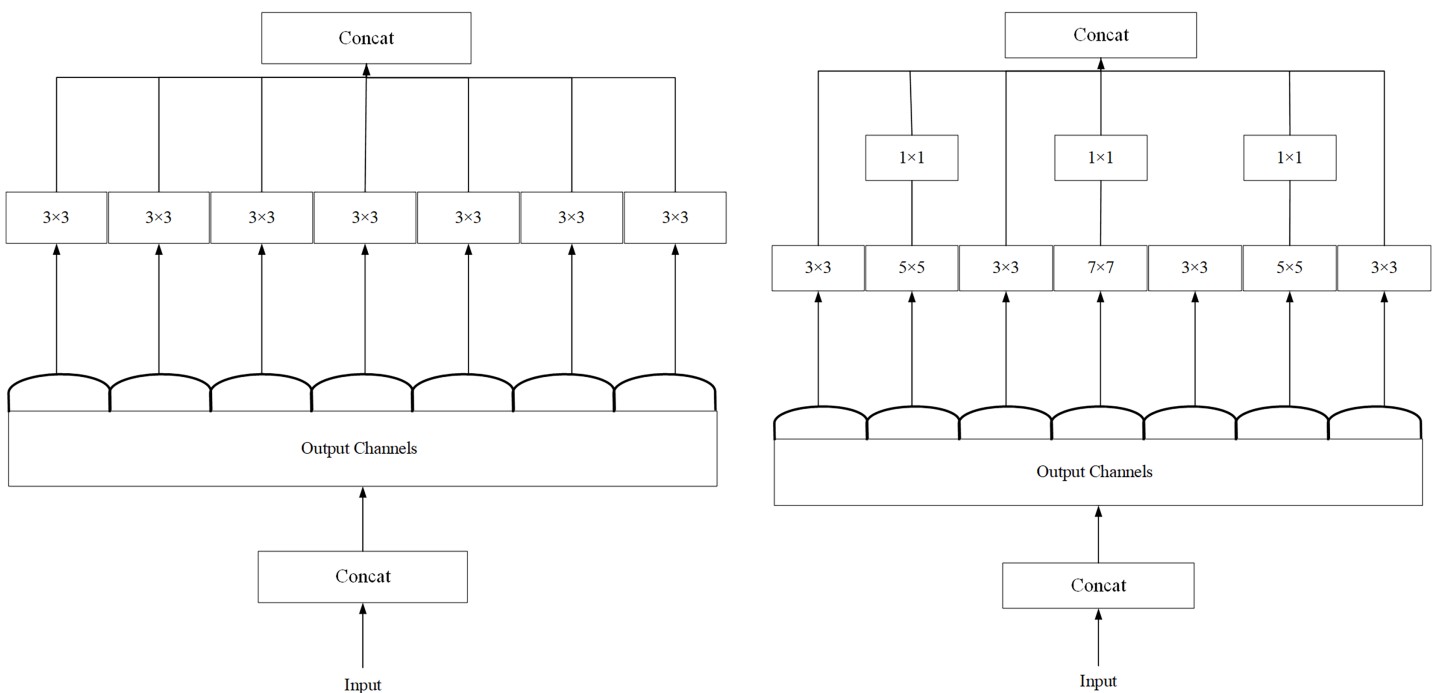

**Figure 5  (A) Structural diagram of the Xception module. (B) Structure of the ED_Xception module.**

## Activation functions

In order to further ensure the stability of the model, we refer to the characteristics of the activation functions of ReLU (*Chen et al., 2020*) and ReLU6 (*Mansuri et al., 2022*), optimize on the swish (*Sudharsan & Ganesh, 2022*) activation function to obtain a new activation function swish6, and apply swish6 to the ophthalmic image classification task,

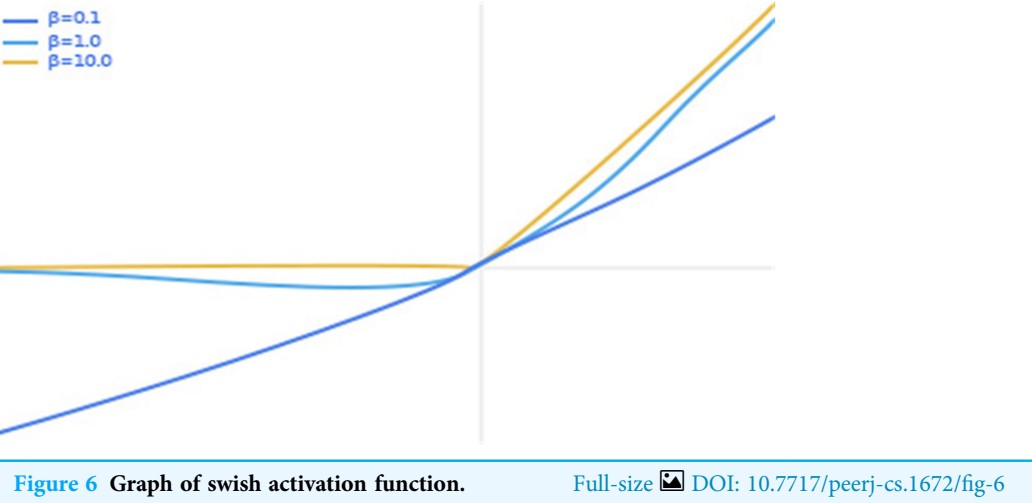

**Figure 6 Graph of swish activation function.**

and obtain better classification results. It is well known that the swish activation function formula is shown in Eq. (2).

$$F(x) = x \cdot Sigmoid(\beta x) \tag{2}$$

where x represents the input pixel value and β is a constant or trainable parameter. The swish activation function is shown in Fig. 6, with separate plots of the swish activation function for values of β of 0.1, 1.0 and 10.0.

We can see from Fig. 6 that as the input value increases, the output value increases. However, when the input value exceeds a certain value, there is a partial gradient explosion in the output value. In view of this, we set a threshold in the swish activation function so that the output value y is fixed at 6 when the input value is greater than 6. The thinking comes from ReLU6 as an improvement on ReLU. Finally, our activation function swish6 is shown in Eq. (3).

$$y(x) = \begin{matrix} x \cdot sigmoid(\beta x) & x < 6 \\ 6 & x > 6 \end{matrix} \tag{3}$$

## Structure of the ED_Xception model

In order to provide a better description of the overall structure of our model, we have produced Table 1 and interpreted the ED-Net network model in detail by Layer, Intput Size, Output Size, Kernel Size, Stride and Padding. As can be seen from Table 1, the structure of our ED-Net network model consists of one ED_Conv, two ED_Resnet, three ED_Xception and one ED-Linear. The input image is taken from a 3 × 3 feature map and the ED-Net network model eventually outputs a seven-class output probability.

To further visualise the structure of the ED-Net network model more intuitively, we visualised the ED-Net output as shown in Fig. 7. We have substituted different colours for

**Table 1 ED-Net network structure table.**

| Model | Layer | Intput size | Output size | Kernel size | Stride/Padding |
|---|---|---|---|---|---|
| ED_Conv | Conv/LBN/swish6 | 3 | 64 | 7 × 7 | 2/3 |
| ED_Resnet | ConV/LBN | 64 | 32 | 1 × 1 | 1 |
| | Dep-ConV/LBN | 32 | 32 | 3 × 3 | 1/1 |
| | ConV/LBN/swish6 | 32 | 64 | 1 × 1 | 1 |
| ED_Xception | ConV/LBN | 64 | 64 | 3 × 3 | 1/1 |
| | ConV/LBN | 64 | 64 | 5 × 5 | 1/2 |
| | ConV/LBN | 64 | 64 | 7 × 7 | 1/3 |
| | ConV/LBN/swish6 | 64 | 128 | 1 × 1 | 1 |
| ED_Xception | ConV/LBN | 128 | 128 | 3 × 3 | 1/1 |
| | ConV/LBN | 128 | 128 | 5 × 5 | 1/2 |
| | ConV/LBN | 128 | 128 | 7 × 7 | 1/3 |
| | ConV/LBN/swish6 | 128 | 256 | 1 × 1 | 1 |
| ED_Resnet | ConV/LBN | 256 | 128 | 1 × 1 | 1 |
| | Dep-ConV/LBN | 128 | 128 | 3 × 3 | 1/1 |
| | ConV/LBN/swish6 | 128 | 256 | 1 × 1 | 1 |
| ED_Xception | ConV/LBN | 256 | 256 | 3 × 3 | 1/1 |
| | ConV/LBN | 256 | 256 | 5 × 5 | 1/2 |
| | ConV/LBN | 256 | 256 | 7 × 7 | 1/3 |
| | ConV/LBN/swish6 | 256 | 512 | 1 × 1 | 1 |
| ED-Linear | Global_pool/Linear | 512 | 7 | | |

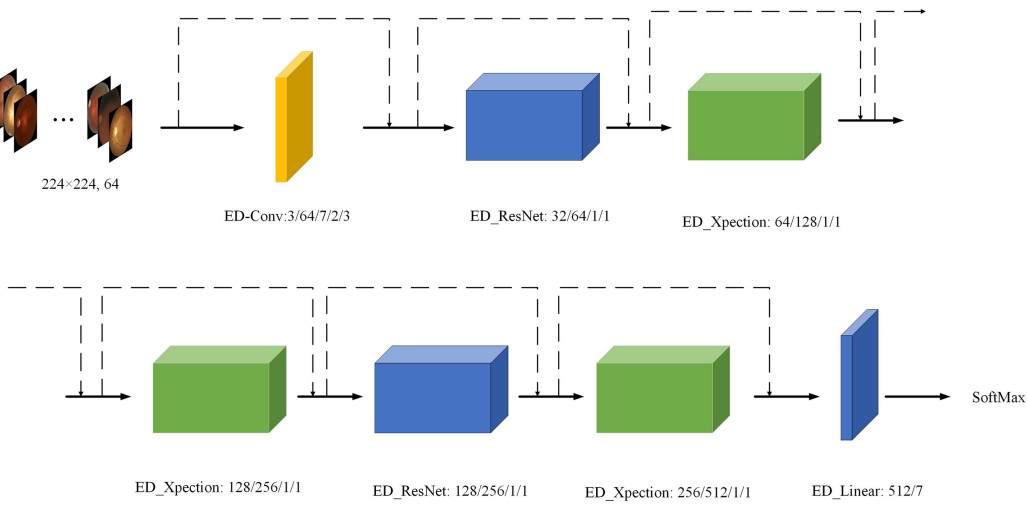

**Figure 7 Visualisation of the ED-Net network.** Eye images credit: Kaggle, © Guna Venkat Doddi.

different models, while the input value is a 224 × 224 sized, three-channel image, and the final output value of each module is annotated in the figure as the input value for the next module.

**Table 2 Data set details table.**

| Category | Train | Test |
|---|---|---|
| Bulging_Eyes | 24 | 6 |
| Cataracts | 813 | 272 |
| Crossed_Eyes | 131 | 43 |
| Diabetic_retinopathy | 823 | 275 |
| Glaucoma | 815 | 274 |
| Uveitis | 40 | 10 |
| Normal | 805 | 269 |
| Total (4,600) | 3,451 | 1,149 |

# EXPERIMENTAL

## Datasets

Our experimental dataset comes from the public dataset of the Kaggle Competition (https://www.kaggle.com/datasets/kondwani/eye-disease-dataset; https://www.kaggle.com/datasets/gunavenkatdoddi/eye-diseases-classification) and Shenzhen Aier Eye Hospital Affiliated to Jinan University, our dataset is divided into a total of seven categories, namely: Bulging_Eyes, There were 4,600 images in our dataset, 3,451 in the training set and 1,149 in the test set, and the details are shown in Table 2. The horizontal and vertical resolutions of the images are 96 dpi and the bit depth is 24 bits.

## Experimental environment

We used Ubuntu 20.04.4 LTS 64-bit operating system and NVIDIA 3080Ti 12G video memory for our experiments. We used an input size of $224 \times 224$ per image, a batch size of 64 for each input model, a uniform algorithm optimizer set to SGD (stochastic gradient descent) (*Woodworth et al., 2020*), an initial learning rate of 0.01, and a learning rate of 95% of the original for each iteration, an epoch of 100 for the number of iterations, a momentum of 0.9, and a weight decay of 0. 100, momentum was set to 0.9, weight decay was set to 0.0004, and the loss function was Cross Entropy Loss Function (CELF) (*Ho & Wookey, 2019*).

## Comparison of experimental data

To further verify the validity of this article, we produced Tables 3–6. we compared the algorithms with classical ones (Vgg16 (*Sitaula & Hossain, 2021*), Resnet50 (*Wen, Li & Gao, 2020*), Densenet121 (*Zhang et al., 2019*), ResNext_34×4d-50 (*Zhou, Zhao & Wu, 2021*), ShuffleNetV2 (*Vu, Le & Wang, 2022*) and Mobilenetv3_ large (*Chen et al., 2022*)), advanced algorithms (Conformer (*Guo et al., 2021*), RepMLP_B224 (*Ding et al., 2021a*), RepVGG_D2se (*Ding et al., 2021b*), ConvMixer (*Ng et al., 2022*), and Hornet-L-GF (*Rao et al., 2022*)), transformer algorithms (DeiT-base (*Touvron et al., 2021*), PoolFormer_M48 (*Yu et al., 2022*), SVT_large (*Fan et al., 2022*), EfficientFormer-l7 (*Li et al., 2022b*) and MViTv2_large (*Li et al., 2022a*)) and similar algorithms for experimental comparison.

**Table 3 Comparison with classical algorithms.**

| Model | Acc (%) | FLOPs (G) | Param# (M) |
|---|---|---|---|
| Vgg16 (2014) | 26.36 | 15.5 | 138.36 |
| Resnet50 (2015) | 79.26 | 1.31 | 23.52 |
| Densenet121 (2016) | 88.96 | 2.88 | 7.98 |
| ResNext_34×4d-50 (2017) | 78.80 | 4.27 | 25.03 |
| ShuffleNetV2 (2018) | 31.20 | 0.149 | 2.28 |
| Mobilenetv3_large (2019) | 55.02 | 0.23 | 5.48 |
| ED-Net | 91.66 | 0.16 | 2.68 |

**Table 4 Comparison with advanced algorithms.**

| Model | Acc (%) | FLOPs (G) | Param# (M) |
|---|---|---|---|
| Conformer (2021) | 86.23 | 4.90 | 23.52 |
| RepMLP_B224 (2021) | 50.22 | 6.71 | 68.24 |
| RepVGG_D2se (2021) | 91.20 | 36.56 | 133.33 |
| ConvMixer (2022) | 89.37 | 19.62 | 21.11 |
| Hornet-L-GF (2022) | 93.02 | 34.58 | 196.29 |
| ED-Net | 91.66 | 0.16 | 2.68 |

As can be seen from Table 3, our algorithm ED-Net is more accurate than all the classical algorithms, while the floating point operations (FLOPs) are all lower than the others except for being on par with ShuffleNetV2. In terms of Param (parameter), our algorithm is 51.6 times lower than Vgg16 and on par with ShuffleNetV2. This indicates that our algorithm is more suitable for application to the ophthalmic disease classification task than the classical algorithm.

As can be seen from Table 4, ours has a high accuracy rate compared to advanced CNN algorithms in terms of accuracy, although our accuracy rate is lower compared to the Hornet-L-GF algorithm and equal to ours compared to the RepVGG_D2se algorithm. Also, on FLOPs, our algorithm ED-Net is 216.1 times less accurate compared to Hornet-L-GF and 228.5 times less accurate compared to RepVGG_D2se; on Param, our algorithm is 73.2 times less accurate compared to Hornet-L-GF and 49.8 times less accurate compared to RepVGG_D2se. Thus, we further demonstrate that our algorithm ED-Net is more suitable for ophthalmic disease classification tasks than advanced CNN algorithms.

It is well known that the transformer algorithm has obtained many superior results in numerous fields, therefore, we compared the transformer algorithm in the course of the algorithm. As can be seen from Table 5, the transformer algorithm performed better overall on the ophthalmic disease classification task, with the highest accuracy being the MViTv2_large algorithm, but the high FLOPs and Param of the MViTv2_large algorithm indicate that the complexity of the algorithm is high and is not suitable for application to the ophthalmic disease classification task. deiT-base, PoolFormer_M48, SVT_large and

**Table 5 Comparison with transformer algorithms.**

| Model | Acc (%) | FLOPs (G) | Param# (M) |
|---|---|---|---|
| DeiT-base (2021) | 88.34 | 16.86 | 86.57 |
| PoolFormer_M48 (2021) | 87.06 | 11.80 | 73.47 |
| SVT_large (2021) | 87.68 | 14.82 | 99.27 |
| EfficientFormer-l7 (2022) | 89.03 | 10.16 | 82.23 |
| MViTv2_large (2022) | 94.20 | 42.10 | 217.99 |
| ED-Net | 91.66 | 0.16 | 2.68 |

**Table 6 Comparison with similar algorithms.**

| Model | Acc (%) | FLOPs (G) | Param# (M) |
|---|---|---|---|
| Literature (*Ting et al., 2019*) | 87.20 | 1.11 | 7.14 |
| Literature (*Frame et al., 1998*) | 88.12 | 0.12 | 0.46 |
| ED-Net | 91.66 | 0.16 | 2.68 |

EfficientFormer-l7 algorithms are all lower in accuracy than our algorithm ED-Net, while our algorithm is still significantly lower than the other transformer algorithms in terms of FLOPs and Param. Therefore, our algorithm ED-Net is more suitable for ophthalmic disease classification tasks than the transformer algorithm.

To further validate the effectiveness of our algorithm, we further conducted experimental comparisons with similar algorithms for ophthalmic diseases, while our experimental code was taken from the two best performing algorithms in the kaggle competition, code from literature (*Yousrry, 2022*) and literature (*Fakher, 2021*), respectively. We can see from Table 6 that our algorithm ED-Net still performs the best in terms of accuracy, 4.46% higher compared to *Yousrry (2022)* and 3.54% higher compared to literature (*Fakher, 2021*), while our algorithm significantly outperforms literature (*Yousrry, 2022*) and is on par with literature (*Fakher, 2021*) in terms of FLOPs and Param. Therefore, we further validate that our algorithm ED-Net has a better performance and is suitable for application in ophthalmic disease classification tasks.

## Visualisation effects

To further verify the validity of the data of the algorithm in this article, we visualized the process of ED-Net's accuracy and loss change. It can be seen from Fig. 8 that the ED-Net network model is stable in the network region at 100 epochs of iterations, which proves that we set a good number of iterations. In addition, as the number of iterations keeps increasing, the accuracy of the ED-Net network model keeps increasing and the loss keeps decreasing, therefore, the ED-Net network training process is normal, which indicates that our ED-Net network model is set up more reasonably. Therefore, the ED-Net network model we set up was reasonable in the training and testing phases, further validating the reliability of our data.

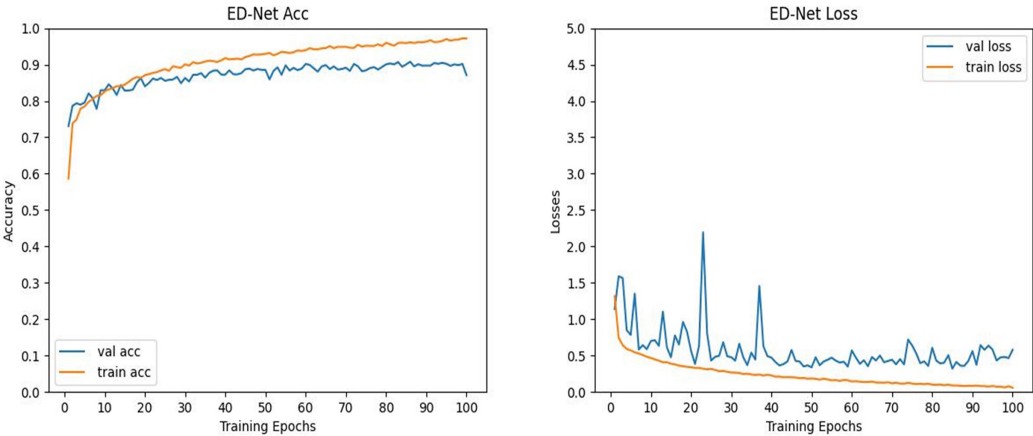

**Figure 8** (Left) Plot of the evolution of the accuracy curve of the ED-Net network. (Right) Variation of ED-Net network loss curve.

## CONCLUSION

With the changing environment of people's eyes and the importance of the eyes to human life, the treatment of ophthalmic diseases becomes crucial. We propose a new ophthalmic disease classification method, ED-Net, based on the residual module of the ResNet network and the ED_Resnet model, and the ED_Xception model, based on the Xception network module. The ED-Net network model is obtained by stacking the ResNet and ED_Xception modules, and the effectiveness of the ED-Net network model is demonstrated in the course of experiments, further helping doctors to contribute to the diagnosis of ophthalmic diseases. In the future, the use of deep learning methods for the treatment of various diseases will be crucial.

### Funding

The study was funded by the Science and Technology Innovation Program of Hunan Province (Grant No. 2021JJ30045). The funders had no role in study design, data collection and analysis, decision to publish, or preparation of the manuscript.

### Grant Disclosures

The following grant information was disclosed by the authors:
Science and Technology Innovation Program of Hunan Province: 2021JJ30045.

### Competing Interests

The authors declare that they have no competing interests.

### Author Contributions

- Fangyuan Liu conceived and designed the experiments, performed the experiments, performed the computation work, prepared figures and/or tables, and approved the final draft.

- Bo Qin analyzed the data, performed the computation work, authored or reviewed drafts of the article, and approved the final draft.
- Fengqi Jiang performed the computation work, prepared figures and/or tables, and approved the final draft.

## Data Availability

The datasets used are available at Kaggle:

- https://www.kaggle.com/datasets/kondwani/eye-disease-dataset (Kondwani);

- https://www.kaggle.com/datasets/gunavenkatdoddi/eye-diseases-classification (guna venkat doddi).

## Supplemental Information

Supplemental information for this article can be found online at http://dx.doi.org/10.7717/peerj-cs.1672#supplemental-information.

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
