# Peer review of "Eye Disease Net: an algorithmic model for rapid diagnosis of diseases"

_PeerJ Computer Science, doi:10.7717/peerj-cs.1672_

## Round 0.1 · original submission · Minor Revisions

Hi,

Kindly improve the quality based on reviewers comments.

Reviewer 1 ·

Basic reporting

rather or relatively good

Experimental design

rather or relatively good

Validity of the findings

rather or relatively good

Additional comments

The authors propose a method for classifying diseases, and we found that the paper is characterized by the following points by reading the whole paper:
(1) The overall logic of the paper is strong, and there are more comparison experiments. First, experiments are compared with some classical algorithms, such as Vgg16, Resnet50, Densenet121, ResNext_34×4d-50, ShuffleNetV2, Mobilenetv3_large; and after that experiments are compared with some very advanced algorithms, such as Conformer, the RepMLP_B224, RepVGG_D2se, ConvMixer, Hornet-L-GF. It is very much appreciated because the algorithms are stated very accurately, for example, Mobilenetv3 is directly written as Mobilenetv3_large which is a very rigorous academic habit.
(2) The authors optimized on Xception and proposed ED_Xception model, is it reasonable to use "ED_Xception model"? Is it reasonable to use "ED_Xception model"? Should it be changed to "ED_Xception module"?
(3) Figure 6: I feel that the quality of the drawing needs to be improved.
(4)Suggested revision and acceptance.

Cite this review as

·

Basic reporting

Dear author,
Thank you for submitting your manuscript to our journal. I have carefully reviewed your paper and would like to provide you with some feedback on my findings. The paper presents a new method for ophthalmic disease classification, called ED_Net. It is a topic of interest to the researchers in the related areas, and the findings are of considerable interest. A few minor revisions are list below. Therefore, I believe that your paper requires “minor revisions” at this point.
1. Abstract: While the abstract provides a brief overview of the study, it does not adequately summarize the main findings of the paper. We suggest that you revise the abstract to provide a clearer summary of your results.
2. Introduction: The introduction section does not provide sufficient background information or context to help readers understand the significance of your research. I suggest that you consider adding more detail to help readers better understand the importance of your work.
3.Clarity and Accuracy: I noticed the figures were unclear. For example, it is not clear that βin Figure 6. Please review the manuscript carefully and make the necessary corrections to ensure that your study’s presentation is clear.
4. Formatting issues: The list of Equation is not on our style. It is close but not completely correct. Before submitting a revision be sure that your material is prepared and formatted.
5.References: There were some minor issues with the references. Please check the references carefully and make the necessary corrections. For example, reference in lines 32 and 33.
6.Future Research: The manuscript does not provide any direction no future research or the next steps following this study’s completion. Please consider providing some guidance in this area.

Experimental design

No comment

Validity of the findings

It remains unclear what the potential value of your study is. I suggest that you articulate this aspect more clearly and persuasively in the manuscript.

Additional comments

Methodology and Organization: The methodology is not presented in sufficient detail, and the organization could be improved to enhance the readability and clarity of the paper.

---

## Round 0.2 · accepted · Accept

Thanks for addressing the reviewer's comments.

Reviewer 1 ·

Basic reporting

No problem. It's better.

Experimental design

No problem. It's better.

Validity of the findings

No problem. It's better.

Additional comments

Read the original and returned content, as well as the code. Receivable, it's a very nice behavior to be able to disclose the code, I hope it stays.

Cite this review as

·

Basic reporting

Dear author,
Thank you for submitting your manuscript to our journal. The paper has certain novelty and advantages for this field research work. It presents a new method for ophthalmic disease classification, called ED_Net. This is a carefully done study and the findings are of considerable interest. Therefore, we believe that your paper requires “accept” at this point.
In conclusion, we believe that your has the potential to make a good contribution to our journal. We appreciate your hard work and look forward to receiving your revised manuscript.

Experimental design

no comment

Validity of the findings

no comment

Additional comments

no comment